# Relationships of *Brassica* Seed Physical Characteristics with Germination Performance and Plant Blindness

Pedro Bello [ID] and Kent J. Bradford *[ID]

Seed Biotechnology Center, Department of Plant Sciences, University of California, Davis, CA 95616, USA; pbello@ucdavis.edu
* Correspondence: kjbradford@ucdavis.edu

**Abstract:** *Brassica oleracea* is an important crop species that at early growth stages may exhibit failure of the apical growing point, an abnormality called "blindness". The occurrence of blindness is promoted by exposure to low temperatures during imbibition and germination, but the causes of sensitivity to such conditions are unknown. We combined three analytical seed technology instruments to explore seed physical properties that are highly correlated with quality parameters and might be used directly for grading or sorting seed lots into subpopulations varying in potential susceptibility to blindness. For image analysis, we used the VideometerLab instrument, which can scan 19 wavelengths from ultraviolet to infrared and utilize that information in any combination to potentially identify unique criteria related to seed quality. The iXeed CF Analyzer was utilized to obtain chlorophyll fluorescence values for individual seeds. Chlorophyll contents of many seeds can be used as an indicator of seed maturity, a major contributor to seed quality. Finally, oxygen consumption measurements of individual seeds as obtained with the Q2 instrument are highly correlated with their performance under a wide variety of conditions. Six Brassica seed lots differed in their susceptibility to induction of blindness or loss of viability due to 48 h hydrated incubation at 1.5 °C. Analysis of physical and respiratory parameters identified some measurements that were highly correlated with the occurrence of blindness. Higher chlorophyll content, as detected by the CF-Mobile and certain wavelengths in the Videometer, was associated with greater occurrence of blindness or death following the induction treatment, suggesting that more immature seeds may be susceptible to blindness. Further research is required, but methods to detect and sort such seeds based on physical characteristics appear to be feasible.

**Keywords:** *Brassica oleracea*; blindness; multispectral; chlorophyll content; seed respiration; seed vigor

## 1. Introduction

*Brassica oleracea* is a morphologically diverse species that has been selected and bred for its leaves (cabbage, collards and kale), stems (kohlrabi), flower shoots (broccoli and cauliflower) and buds (Brussels sprouts). During early seedling growth, plants of all of these crops may lose the apical growing point, an abnormality called "blindness", which usually occurs at low incidence but can cause major losses in the field for growers under some conditions. The occurrence of blind *B. oleracea* plants was described already in the 1940s. It is characterized by termination of leaf primordia initiation and disorganization in the shoot apical meristem (SAM) [1]. The occurrence of blindness is promoted by low temperature combined with low light conditions, and seed production conditions can play a role in the seed lot sensitivity as well [2]. Recent studies have confirmed that both genetics and seed production environment contribute to the occurrence of blindness [1]. Seed treatments that reduce susceptibility to blindness also have been developed [3]. Early identification of affected plants before transplanting them into the field has not been possible, resulting in high economic losses that can be up to 95% in broccoli under some conditions [2].

Seed production for these crops can be complex due to their indeterminate flowering habit [4]. At a given time during production, these indeterminate species will have immature, mature, over-mature and shattering seeds present simultaneously. Early seed harvest can result in poor seed quality and low germination due to immaturity [5], while delayed harvest may sacrifice up to 50% of seed yield under adverse conditions [6]. In addition to losses due to shattering, a significant fraction of the seed lot is discarded during seed processing due to the removal of smaller, immature seeds. This variability in maturity levels can impact seed lot quality, as immature seeds will lose vigor and viability at a faster rate than mature seeds [5]. These problems can prevent sale of seed lots that do not reach minimum germination levels, with economic losses from discarded lots.

Here we combine three analytical seed technology instruments to explore seed physical and physiological properties that could be highly correlated with quality parameters and potentially used for grading or sorting seed lots to remove lower quality subpopulations. For image analysis, we used the VideometerLab instrument, which can scan 19 wavelengths from ultraviolet to infrared and utilize that information in any combination to measure seed size, detect microbes, classify damage, and potentially identify unique criteria for assessing seed quality [7–11]. The iXeed CF Analyzer was utilized to obtain chlorophyll fluorescence values for individual seeds. Seed chlorophyll contents of many species can be used as an indicator of seed maturity, a major contributor to seed quality [5,12–16]. Finally, oxygen consumption measurements of individual seeds as obtained with the Q2 instrument are highly correlated with their performance under a wide variety of temperature, water potential, hormonal, priming, aging, and other conditions [17–19]. We used these methods to explore the possibility of identifying early indicators of susceptibility to induction of blindness in kohlrabi seeds.

## 2. Materials and Methods

### 2.1. Seed and Plant Materials

Six kohlrabi (*B. oleraceae* L. var. *gongylodes*) seed lots comprised of three F1 varieties (A, B, C) with two lots of each (1 and 2, 3 and 4, 5 and 6, respectively) exhibiting different susceptibilities for blindness were provided by Bejo Zaden (Warmenhuizen, The Netherlands).

### 2.2. Blindness Induction

After initial measurements of physical characteristics of dry seeds, a blindness induction treatment was performed on the seeds preceding respiration measurements. We adapted a published protocol that demonstrated the ability of low temperature treatments to cause shoot apical meristem arrest in *Brassica oleracea* seedlings [1]. Seeds were imbibed in microtiter plate wells in 60 uL of water and incubated at 1.5 °C in a foil-covered incubator (Benchmark IS-1010R placed in a 4 °C room) for 48 h in darkness and then transferred to respiration tests, maintaining individual seed positioning and identities from previous seed imaging throughout.

### 2.3. Physical Characteristics Measurements

Chlorophyll content. The iXeed CF Analyzer (Figure 1; CF-Mobile; SeQso B.V., The Netherlands) was utilized for chlorophyll fluorescence (CF) measurements [5]. A set of 46–48 seeds per lot were placed on blue metal trays with proper-sized pockets, organized in six rows by eight columns, corresponding to the plate layout and capacity for seed respiration measurements. Three measurements were captured on each tray and seed parameters registered by the CF software were recorded and exported to Microsoft Excel (version 16). The average and standard deviation of CF level and CF size per seed were calculated and combined with parameters gathered subsequently. CF information was captured for a total of 174 seeds for each lot. The parameters derived from the CF-Analyzer data are defined in Table 1.

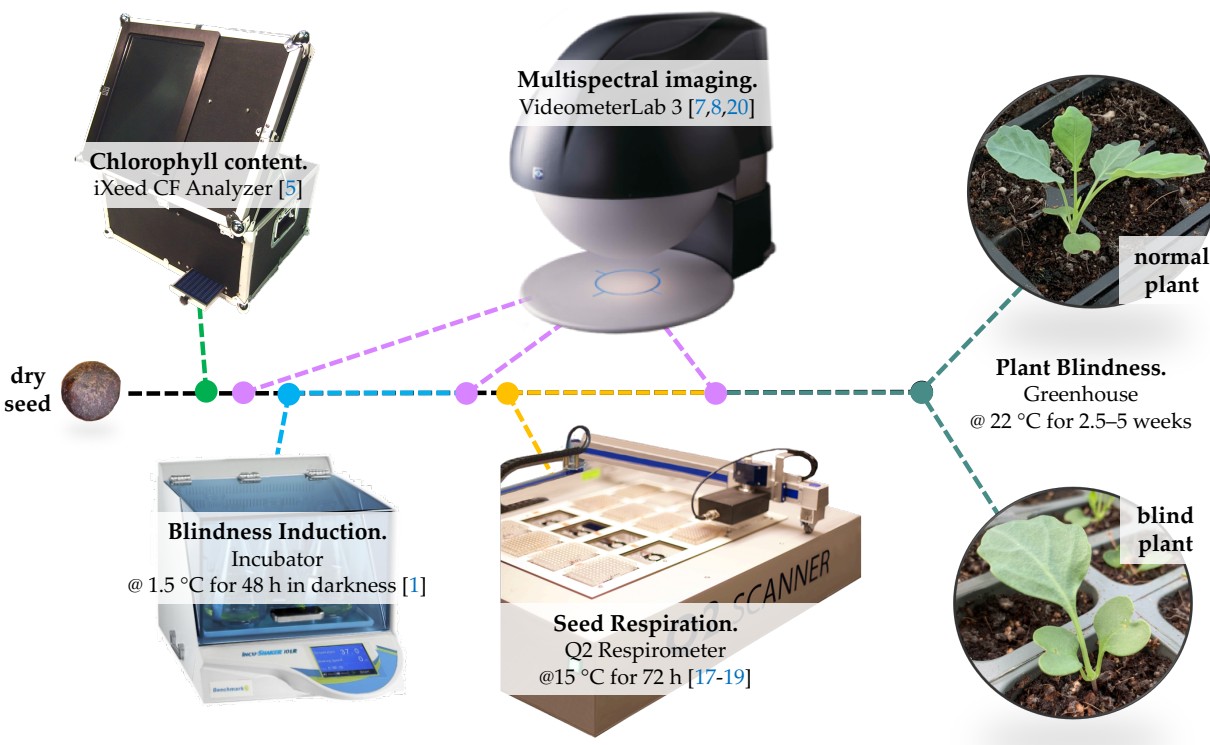

**Figure 1.** Visual workflow for every seed measured in the study. Starting with chlorophyll content and multispectral imaging taken of the dry seeds (left), followed by the blindness induction treatment, additional multispectral imaging, then seed respiration measurements, additional multispectral data collection and finally seeds were transferred to the greenhouse for growth and plant blindness evaluation.

Multispectral imaging. The VideometerLab instrument (Videometer A/S, Herlev, Denmark) was used for multispectral analyses [7,8,20]. The instrument is equipped with a camera inside an integrating sphere along with diodes that emit light at the following 19 wavelengths: 375, 405, 435, 450, 470, 505, 525, 570, 590, 630, 645, 660, 700, 780, 850, 870, 890, 940 and 970 nm. The same 174 seeds initially scanned for CF information were imaged and analyzed in the VideometerLab instrument for each lot. Multispectral pictures were taken and grouped for each 46–48 seeds, respecting the CF measurements positioning. These seeds were placed in coded 6 (rows) by 8 (columns) cells with each seed placed slightly lower than the previous one, aiding the Videometer software sequence numbering. Over two experimental repetitions, multispectral images of the same seed were captured in different stages of the experiment (see Figure 1): (1) dry seed (all 174 seeds per lot); (2) after blindness induction treatment (110 seeds per lot); and (3) after 72 h at 15 °C for seed respiration measurements (96 seeds per lot) where these seeds were rapidly transferred to marked microtiter plate lids and Videometer images were acquired to quantify seedling area, respecting the seed positioning from the Q2 equipment. A series of up to eight pictures was taken per Q2 plate to avoid overlapping tissues and data were combined together in the BLOB (Binary Large Objects) collection. Selected parameters derived from the VideometerLab data are defined in Table 1.

**Table 1.** Parameter Definitions.

| Type | Parameter | Description |
|---|---|---|
| Multispectral | Area | Average or individual projected area (mm$^2$) calculated for the seed. |
| | CIELab | CIELab refers to a color space defined by the International Commission on Illumination (CIE); it expresses color as three numerical values: L for lightness from black (0) to white (100), A from green ($-$) to red ($+$) and B from blue ($-$) to yellow ($+$). |
| | Saturation | Average amount of pixels that exceed a maximum value of brightness in the image. |
| | Hue | Average angular position related to a color space coordinate enclosing all colors. |
| | 395–970 nm | Average reflectance of specific wavelengths (in nanometers) for individual seeds. The specified wavelength followed by "SD" refers to the average standard deviation or pixel reflectance variation for individual seeds. |
| | Tissue Area | Area of uncoated or visible tissue present in individual seeds, quantified by the number of pixels. |
| | White Spots | Area of white coloration in seed coats as a percentage of the individual seed area. |
| Chlorophyll Fluorescence | CF Value | Average or individual chlorophyll fluorescence measured for the seed (total seed fluorescence divided by fluorescence size). |
| | CF Size | Average or individual calculated size (mm$^2$) of the CF area in the seed (sum of the pixels that have a CF level above a threshold, and converted to mm$^2$). |
| Seed Respiration | R75.Time, R50.Time and R25.Time | Time in hours for individual seeds to deplete oxygen in vials to 75, 50 and 25%, respectively. |
| | R75, R50, R25 POD curves | Cumulative population oxygen depletion (POD) time course plotting the percentage of seeds depleting the oxygen level to 75, 50 or 25% of the initial, respectively, at each time. |
| | R50 (50) | Time in hours to 50% level reached on the R50 POD curve. |
| | R75.Final, R50.Final and R25.Final | Final percentages of the R75, R50 and R25 POD curves, respectively, of the initial value. |
| | Final-O$_2$ | Final oxygen concentration in the vials after 72 h of imbibition. |
| Greenhouse Plant Evaluation | Plant Blindness Score | Plants were ranked as dead, normal or blind. Blind plants included: plants without shoot apical meristem [SAM]; plants with needle shaped (first) leaf; plants with funnel shaped (first) leaf; plants without SAM, but with lateral branches; plants with SAM and lateral shoots that are needle- or funnel-shaped or otherwise malformed; plants with oversized first leaf but lacking SAM; plants with abnormal branch architecture. |

### 2.4. Seed Respiration Measurements

The Q2 instrument (now called Seed Respiration Analyzer; Figure 1; Fytagoras B.V., Leiden, The Netherlands) measures oxygen consumption (respiration) rates of individual seeds repeatedly during imbibition and germination. Individual seeds were placed into 2 mL screw-cap vials containing 1.45 mL of agar (0.4% *w/v*) and 0.2% Plant Preservative Mixture (PPM$^{TM}$), which were sealed with caps that have a dot of a fluorescent polymer centered on their internal side. The polymer contains a dye that changes its fluorescent properties in response to oxygen concentration [21]. As the seed respires, it depletes the oxygen in the sealed well or vial, which changes the fluorescence intensity of the dye. This change is detected by a light source that shines on the dot and a sensor that measures the fluorescence intensity. A robotic arm sequentially moves the light source/sensor over each well, measuring the oxygen concentration inside the wells. Up to 16 plates of 48 vials can be positioned in the apparatus at a time and automatically measured by the robotic sensor, and the measurements can be repeated frequently to obtain time courses of oxygen consumption activity. Measurements reported here were collected every 30 min. Seeds were transferred individually to 2 mL vials using tweezers after pictures were taken in the CF-Analyzer and the VideometerLab. Sample temperature was controlled to ±0.5 °C using Peltier heating/cooling units and fans. In preliminary tests, 48 non-induced seeds (control) per lot were measured at 20 °C, followed by plant blindness evaluation. Thereafter, all seeds were measured in separate 48-well plates at 15 °C for 50 or 72 h, and placement in the Q2 cells was paired to the Videometer and CF-Mobile codes for each seed when applied. Two experimental repetitions were utilized. The parameters derived from the Q2 data are defined in Table 1.

### 2.5. Plant Blindness Evaluation

After the blindness induction treatment and the respiration measurements, seeds were transplanted to marked trays with numbered cells and placed in a greenhouse at 22 °C and natural light. The day length during plant growth varied from 13 to 15 h in the initial experiment (carried in April through May 2019) and 13 to 11 h during the experiment repetition (September through October 2020). Individual plants grown for 2.5–5 weeks were evaluated and ranked as dead (no seedling emerged), normal or blind plants (Figure 1). The parameters derived from the plant evaluations are defined in Table 1.

### 2.6. Data Analyses

Data analyses for Videometer images were performed using VideometerLab and the Classifier Design Tool (CDT) software version 3.18.11 (Videometer A/S). We separated seeds from the image background using normalized canonical discriminant analysis (nCDA) transformation followed by simple threshold segmentation within the Videometer software. Several multispectral images were used to create the nCDA transformation model with selected areas of images representing the 2 classes: areas of seeds or background. Automatic normalization was performed to maximize the Rayleigh quotient and input data received a preprocessing band normalization (at 645 nm) and output data was centered around the overall mean between the classes and scaled with the two classes showing means at +1 or −1. The actual data were visualized in a scaling between −2 and 2.

A similar approach was used to quantify the visible tissue area after seed respiration measurements, but in this case using the input with multispectral images featuring the seed coat or embryo/seedling tissue area as the two classes to be separated. The input normalization in this case was performed using preprocessing band normalization at 470 nm, while output normalization was similar. The seedling or tissue areas were quantified by numbers of pixels.

Data from the CF-Analyzer, Q2 and plant evaluations were exported or compiled using Microsoft Excel (version 16). The compiled data were then analyzed in R version 4.0.3 using RStudio version 1.2.1335. Type I analysis of variance (ANOVA) was run on models with experiment as random effect. Normality and heteroscedasticity of the data

was visually inspected with histograms and diagnostic plots for all parameters reported using the linear regression analysis (lm) models in R. Tukey Honest Significant Differences were then calculated using the TukeyHSD() function in R and the HSD.test() function of the R package *agricolae* [22].

Boxplots were made using ggplot from the ggplot2 R package [23]. Correlation matrices were calculated using the rcorr.test function from the psych R package [24] and plotted with the corrplot R package [25]. Family-wise error rate was accounted for with adjusted *p*-values for multiple comparisons using the Holm method [26]. Multiple factor analysis (MFA) was performed in R with the FactoMineR package [27] and additional tools from the factoextra package [28]. Only quantitative variables without missing values and statistically correlated with blindness ($p < 0.01$) were used for the MFA. All comparisons mentioned were statistically significant at $p < 0.05$ unless otherwise stated.

## 3. Results

### 3.1. Brassica Blindness: Initial Assessment

All seed lots were first tested for seed respiration and plant growth to identify blindness present in the seed lots and to characterize their vigor prior to the blindness induction treatment. The initial seed respiration test was conducted at 20 °C and all seed lots displayed largely homogeneous and rapid oxygen depletion rates (Supplemental Figure S1). At least 80% of seeds in all lots depleted oxygen in the vials to the 50% level within two days after imbibition and at 72 h most seeds were in anaerobic conditions in the vials. These seeds were then transferred to the greenhouse for plant growth evaluation, and no blind plants were identified after 2.5–5 weeks (data not shown).

To further investigate the blindness potential and susceptibility in all lots, we introduced moderate temperature stress during germination by lowering the temperature to 15 °C in the Q2 test. As expected, oxygen depletion rates were slower for all lots and larger variation in respiratory patterns within lots was also evident (Supplemental Figure S2). The modest temperature stress did not induce blindness, with only one seed showing some blindness symptoms in Variety B, lot number 3. Additionally, a significant number of seeds in most seed lots (except A-2 and B-4) did not reduce the oxygen within vials to the 50% level following imbibition for 50 h. The results demonstrated that little or no blindness was expressed in the seed lots tested under optimal or moderately low temperatures during imbibition and germination.

### 3.2. Physical Characteristics Assessment and Brassica Blindness Induction

We tested whether the CF-Analyzer, the VideometerLab or the Q2 were able to detect differences among lots and the potential presence of blind seeds. Chlorophyll fluorescence measurements were initially captured for all dry seeds (Figure 2). The seed lots within B and C varieties displayed a significant difference between each other ($p < 0.001$) while the lots in Variety A did not. Seed lots 5 (Variety C, 90.7) and 3 (Variety B, 78.4) displayed higher median chlorophyll contents when compared to the other lots (27.9–37.6). Additionally, these two lots had higher CF level variation (Figure 2, CF Level—larger bars/standard deviation on lots 3 and 5) in comparison to all other seed lots, which had more homogeneous low CF levels, although some outliers were present in these lots (Figure 2, black dots on lots 1, 2, 4 and 6). These lots with low median CF levels were not statistically different from each other but were significantly different from lots 3 and 5 (Figure 2, $p < 0.001$). Similar differences among lots were detected for CF size (or area) (Figure 2).

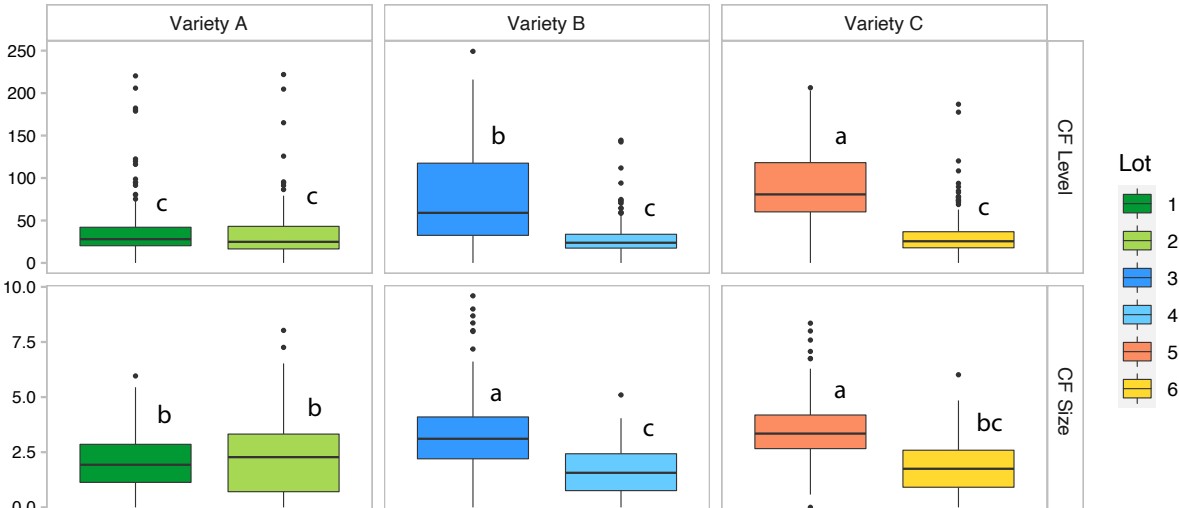

**Figure 2.** CF-Analyzer parameters (see Table 1) across varieties (columns) and seed lots (color-coded). Letters indicate significant differences among all lots for each parameter (rows) as calculated by ANOVA and Tukey HSD.

Multiple seed features were measured using the Videometer (123 parameters in total over all experimental stages, Supplemental Table S1), including seed size, shape, color and multispectral characteristics. Selected features that displayed some relationship with plant blindness, viability, or parameters gathered by the other analytical equipment used here are described in Table 1. Individual data were obtained for 174 seeds (46 seeds in the first repetition and 128 in the second repetition) for each seed lot (Supplemental Table S1). Several captured seed features differed significantly among lots (Figure 3). The calculated seed area was largest for seed lot 2 (Variety A, 3.84) and smallest for seed lot 4 (Variety B, 2.96). Color space (CIELab L, B) and saturation values together with average reflectance at longer wavelengths, such as red (645 nm) and near-infrared (870 nm) showed consistently and significantly higher ($p < 0.001$) values for seed lots 5 (C) and 3 (B) compared to the other lots (Figure 3), as observed for their CF values. A similar result was obtained for the ultraviolet (375 nm) wavelength, but it also included lot 1 ($p < 0.001$) along with the other two high-valued lots. Color and multispectral values for lot 5 also were significantly higher than for other lots ($p < 0.001$) but comparable with lot 1 for reflectance measured at the indigo color (435 nm) and also similar to the two Variety A seed lots (1 and 2) for Hue values. Wavelengths 435 and 645 nm are indications of chlorophyll A and B levels, respectively.

The spectrum reflectance standard deviation within seeds (or pixel variation) was also calculated on an individual seed basis for all wavelengths. This value quantifies the color or spectrum variation of each seed; seeds with a uniform color will display small values while seeds with a diversity of colors or shades will display larger values. Here we show the reflectance standard deviation for the near-infrared (NIR) 970 nm wavelength, which displayed some relationship with plant performance when measured at the dry seed and after blindness induction phases, although the standard deviation for other wavelengths also displayed similar results (Supplemental Figure S3). The calculated standard deviation for the NIR wavelength (970 nm) showed larger median variation (5.75–6.02) for seed lots 3 and 5 with lowest values for lots 6 (4.86) and 2 (4.50) (Figure 3).

The post-blindness-induction (PBI) seed area had a median increase of about 20% for all lots compared to dry seeds, reflecting expansion due to imbibition (Figure 4). The color space parameter CIELab B (blue (−) to yellow (+)) had a median increase of about 50% in most lots with a smaller increase of 32.2% observed in lot 5 (Variety C), which presented the higher ($p < 0.001$) value for CIELab B before induction (Figure 3). Similar relationships were observed for the saturation, hue and 970 nm-SD values (Figure 4), in which the seed lot with the lower initial value had the largest relative increase after blindness induction.

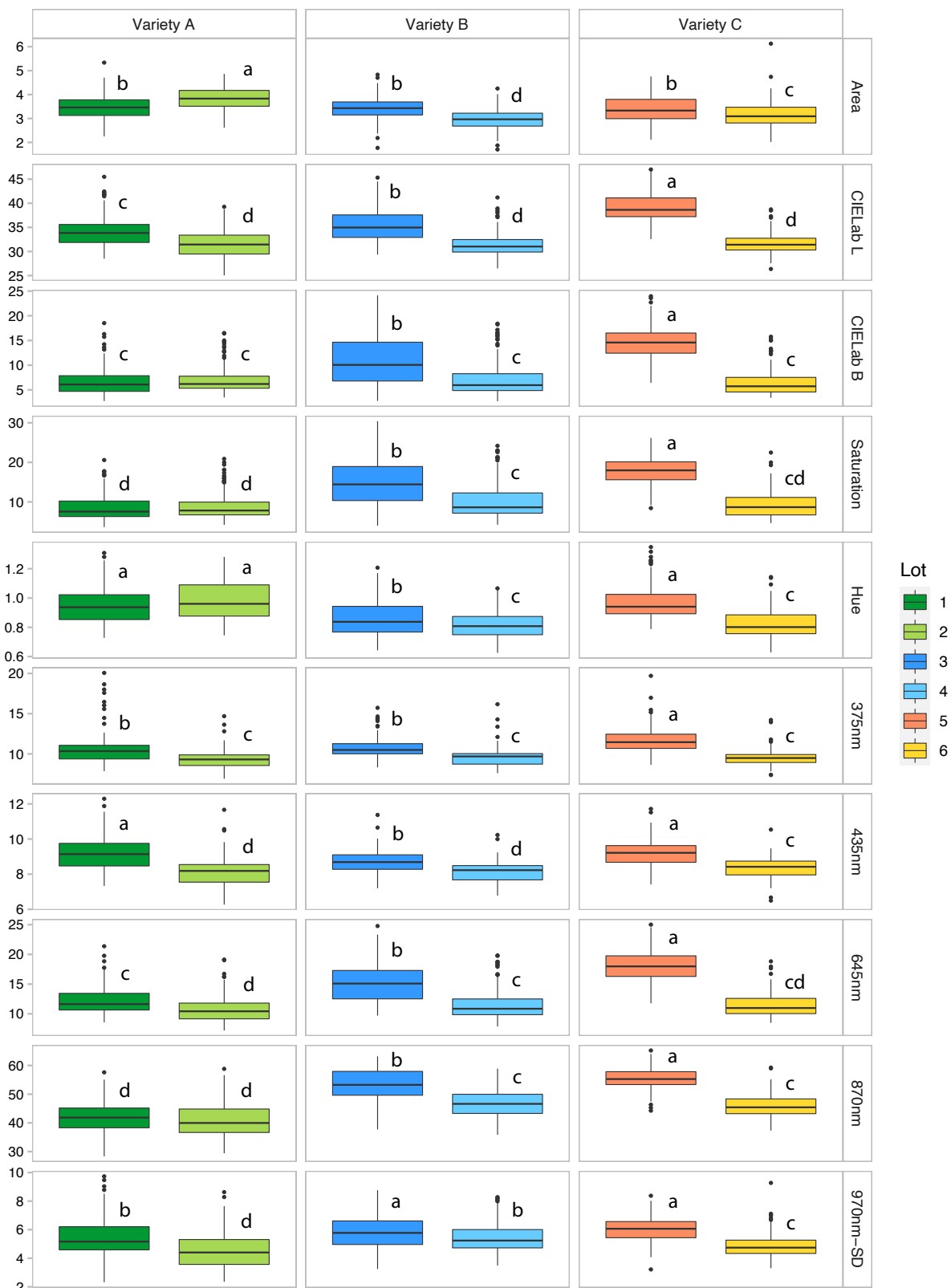

**Figure 3.** Selected Videometer features (Table 1) across varieties (columns) and seed lots (color-coded). Letters indicate significance between all lots for each parameter (rows) as calculated by ANOVA and Tukey HSD.

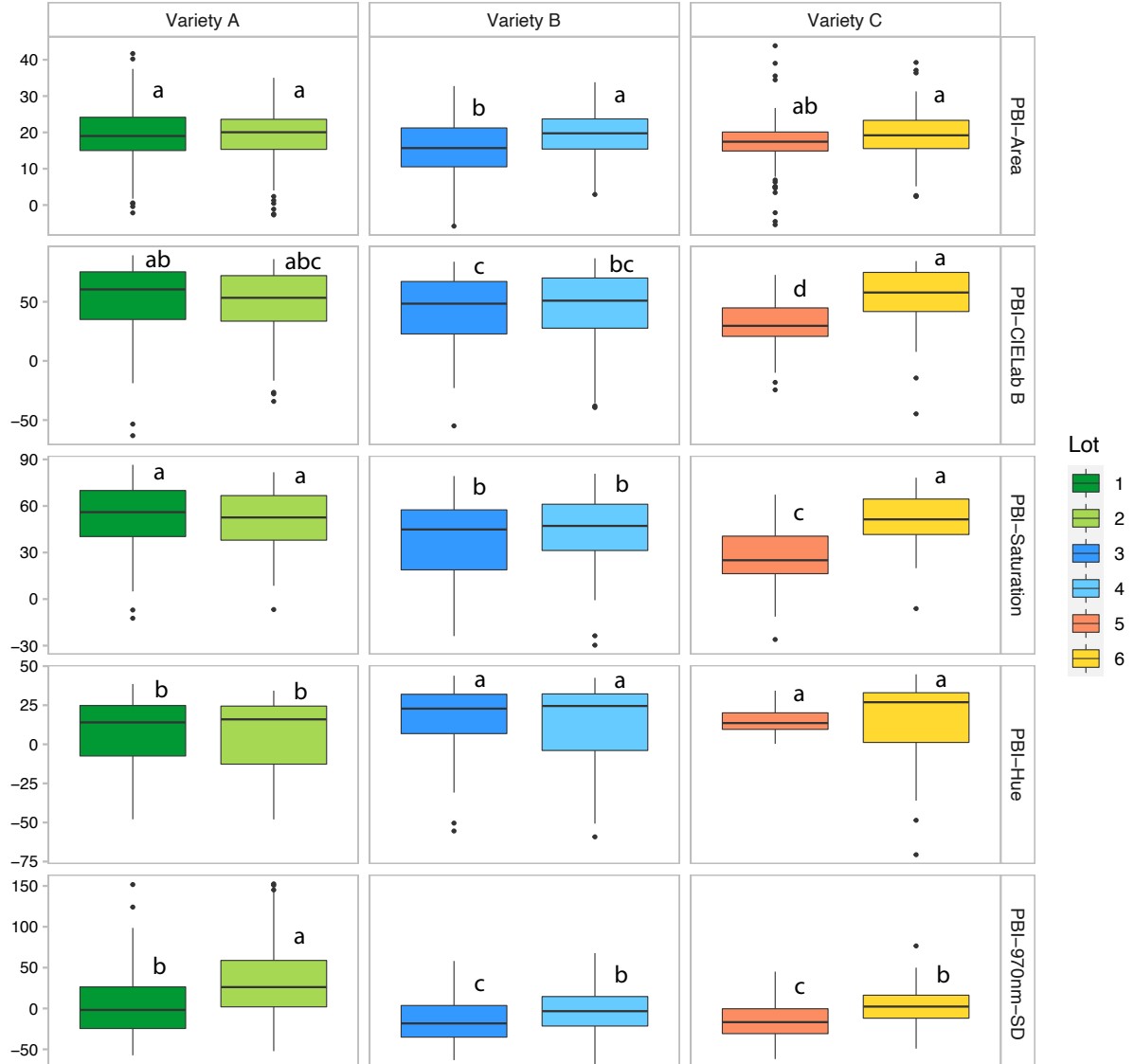

**Figure 4.** Selected VideometerLab features presented as percentage change in the feature levels post-blindness induction (PBI) relative to initial dry seed analyses for the same seeds. Letters indicate significant differences among all lots for each parameter (rows) as calculated by ANOVA and Tukey HSD.

Q2 measurements at 15 °C on the six seed lots after the cold temperature imbibition treatment indicated an overall delay in respiration rates (Figure 5, Supplemental Table S2—bottom section, mean R75 values ranging from 26.2 to 42 h) compared to measurements at a similar temperature on seeds prior to the induction treatment (Supplemental Figure S2, Supplemental Table S2—middle section, mean R75 values ranging from 22.5 to 30.6 h). In addition, a much larger fraction of induced seeds did not consume oxygen after 72 h (Figure 5, Supplemental Table S2—bottom section, e.g., final R75 POD curves values ranging from41 to 98%) compared to non-induced seeds tested earlier for 50 h (Supplemental Figure S2 and Table S2—middle section, e.g., final R75 POD curves ranging at 80–93%). The mean oxygen level at different time points is also a convenient parameter to quantify the respiration profile and variation (Supplemental Table S2, O$_2$ at 48 and 72 h, mean and standard deviation, respectively). Lack of oxygen consumption usually indicates lack of seed viability [19], suggesting that the blindness induction treatment had killed some seeds, as was reported previously regarding this treatment [1].

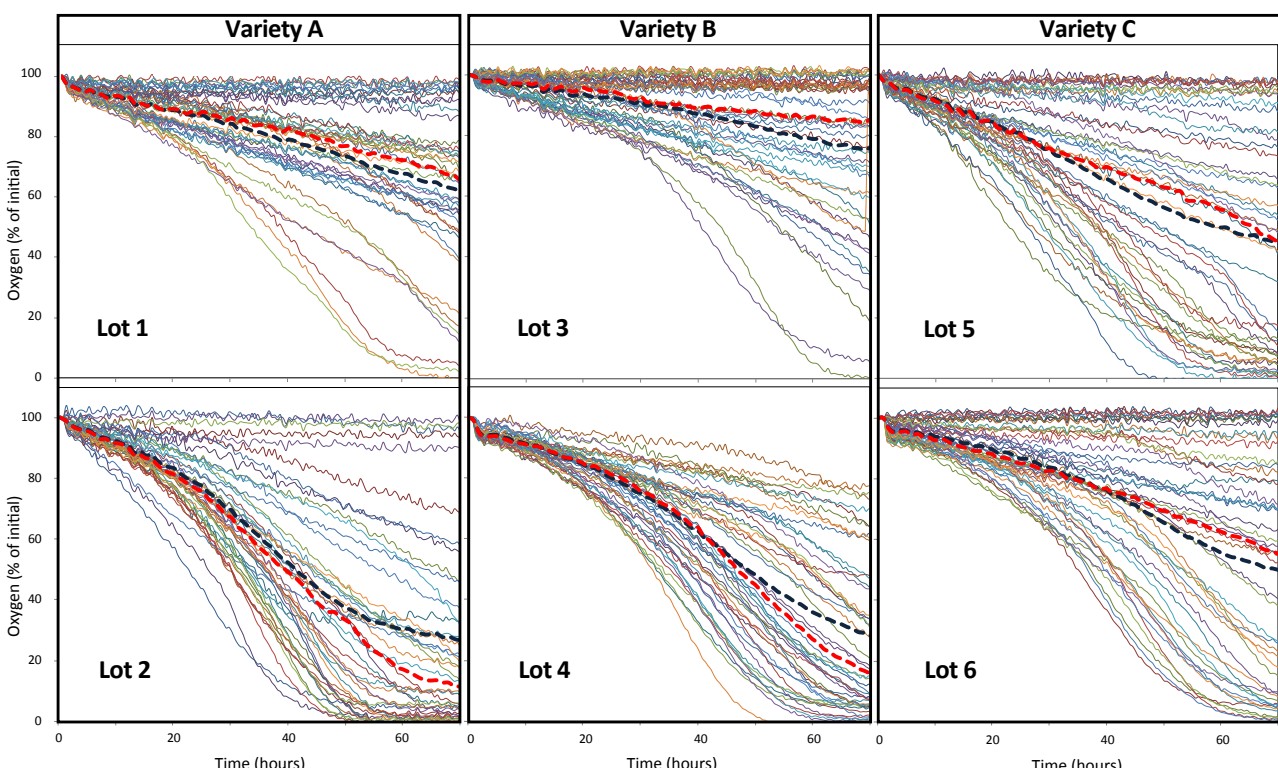

**Figure 5.** Oxygen depletion curves for individual seeds of kohlrabi seed lots tested at 15 °C for 72 h after blindness induction treatment at 1.5 °C for 48 h. Dashed lines represent the median (red) and average (black) oxygen depletion time courses for the entire seed population.

Based on the median oxygen depletion and R75 POD curves (Figure 6), lots 2 (light green) and 4 (light blue) had overall faster median oxygen depletion rates and higher percentages of seeds depleting the oxygen to at least the 75% level. Lot 5 (orange) contained a fraction of seeds respiring even faster than lot 4, but only around 72% of seeds in that lot consumed more than 75% oxygen in the vials, compared to 89 and 100% in lots 2 and 4, respectively. Lots 1 (dark green) and 3 (dark blue) displayed the lowest oxygen consumption (final medians of 65.5 and 84.5% oxygen remaining, respectively) and slower oxygen consumption (2 lowest R75 POD curves), while lot 6 (yellow) performed somewhat better (final median $O_2$ depletion curves of 53.7% and slightly faster R75 POD curves).

Some Q2 parameters were selected for comparison among lots (Figure 7). Lots 1, 3 and 6 displayed the slowest median times to 75% (36.2 to 42 h) and 50% (47.6 to 58.2 h) remaining oxygen levels. Area under the curve parameters (R75.Area and R50.Area) are highly correlated with the time to required to lower the oxygen to the same levels (R75.Time and R50.Time) but add more detailed information regarding the oxygen consumption profiles and shapes of depletion curves. As expected, distributions of areas under the curve for the 75% oxygen remaining level also showed lots 1, 3 and 6 as slower ones (larger area values ranging from 30.95 to 36.44), but at the 50% oxygen level lot 6 displayed a somewhat faster but significant ($p < 0.001$) oxygen consumption compared to lot 3 (Figure 7, R50.Area). Seed lots 2, 4 and 5 usually showed lower median times and area values compared to the slower lots and could be considered significantly ($p < 0.001$) faster respirators in most cases. It is important to point out that the lower the remaining oxygen level chosen to compute these values, the smaller the fraction of seeds that are used to calculate them. In this case at least 40% of the seeds were used to calculate parameters generated based on the 75% oxygen level (lot 3 with lowest final percentage in the R75-POD curve, Figure 6) but a little over 20% of the seeds were used to calculate values based in the 50% oxygen level (slow lots 1 and 3—R50 POD curves, Supplemental Table S2—bottom section). To overcome this issue for less vigorous or stressed lots, the remaining oxygen levels at a particular

time for all seeds can be used. For example, the final oxygen levels in the vials after 72 h (before seeds were transferred to the greenhouse), clearly showed the wide distributions of respiration rates among seeds in these lots after the blindness induction treatment. Seed lots 2 and 4 had relatively homogeneous low median final oxygen levels (19.4 and 22%, respectively), lot 3 displayed an intermediate variance but at the highest final oxygen level (66.9%), while lots 1, 5 and 6 displayed more intermediate final oxygen levels (41.7, 39.4 and 33.1%, respectively) but with large heterogeneity among seeds (Figure 7, Final-$O_2$).

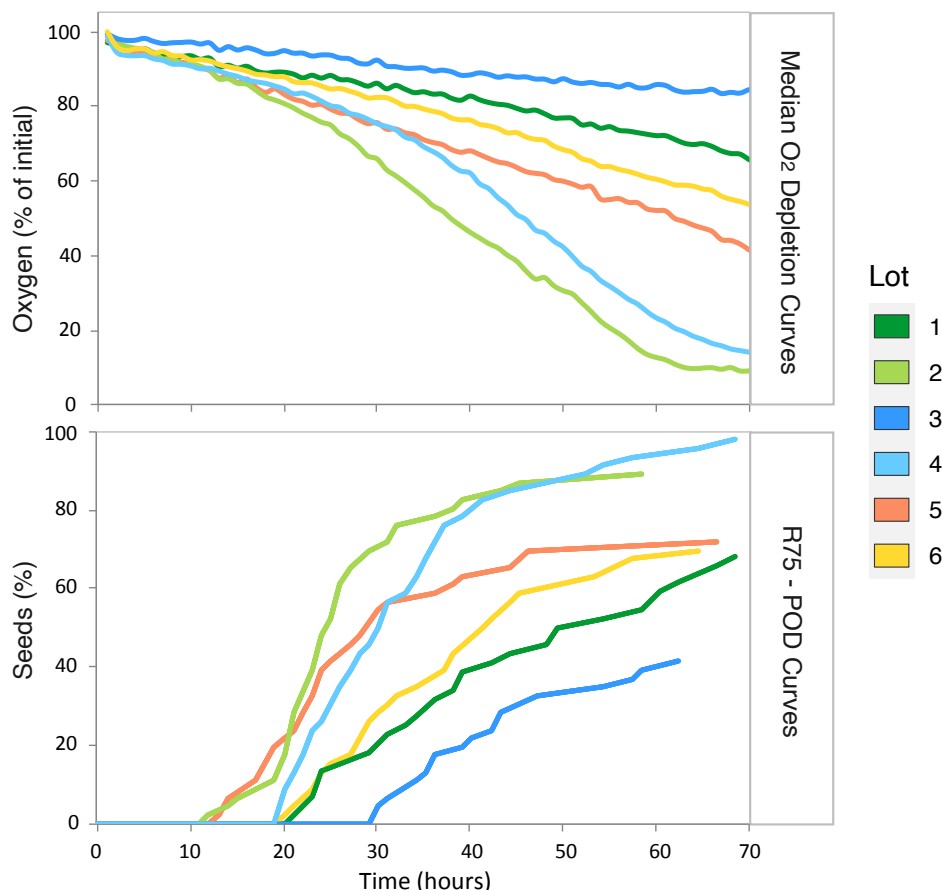

**Figure 6.** Median oxygen depletion curves (top) and R75 POD curves (bottom) of kohlrabi seed lots tested at 15 °C for 72 h after blindness induction treatment at 1.5 °C for 48 h.

Seedling or tissue area measurements from the Videometer using nCDA models (Figure 8—top sections) after seed respiration measurements exhibited significant differences among seed lots. Seed lot 3 had the smallest exposed tissue area (median at 1602 pixels) with little to no seedling tissue visible in a large fraction of the seeds after the respiration measurements (Figure 8—see bottom panels for BLOB collection with seeds marked with dark blue circles compared to seed lot 4 of the same variety marked with light blue circles). Seed lot 5 displayed an intermediate median tissue area (2809 pixels), while lots 1, 2, 4 and 6 had higher median tissue areas at around 4200 pixels (Figure 9). The largest seedling size variation was present in seed lot 1, where a fraction of seeds showed little to no embryo tissue while another fraction displayed the largest seedlings in the study. This is consistent with the large variation among seeds in this lot for Q2-derived values (Figure 7).

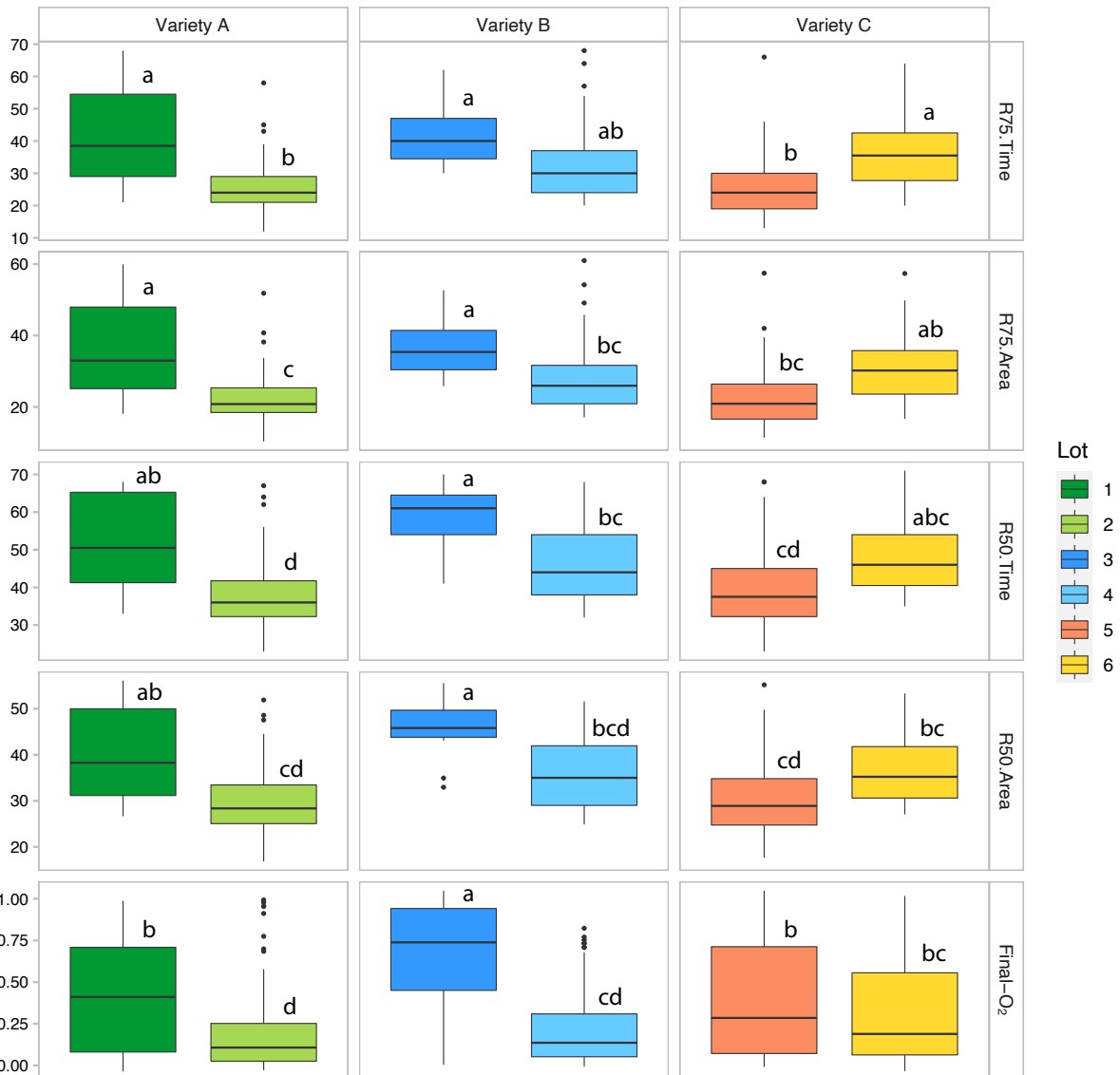

**Figure 7.** Selected Q2 parameters (see Table 1) of seed lots after blindness induction treatment. Letters indicate significance between all lots for each parameter (rows) as calculated by ANOVA and Tukey HSD.

Finally, seeds were transferred to marked trays and placed in the greenhouse at 22 °C for 2.5–5 weeks, when plants were scored as normal, blind or dead (failed to emerge). Lots 2, 4 and 6 had the largest fractions of normal plants (≥80%) while lots 1, 3 and 5 had smaller fractions of normal plants (>62%) (Figure 10). Additionally, lots 3 (19.1%) and 5 (20.9%) had the largest fractions of blind plants, followed by lot 1 (10.9%), lot 4 (7.3%) and lot 6 (2.7%); lot 2 did not display any blind plants after the induction treatment. The percentages of non-viable plants were higher in lots 3 (43.6%), 1 (27.3%) and 5 (26.4%), while lots 4 (12.7%), 6 (10.9%) and 2 (2.7%) exhibited lower percentages of seed death (Figure 10).

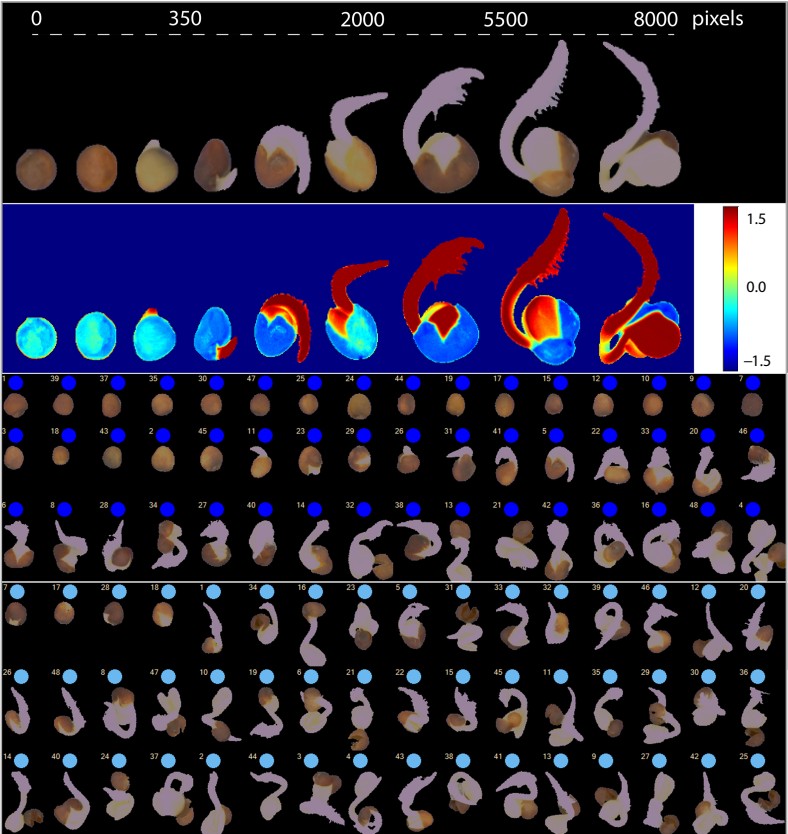

**Figure 8.** Examples of seedling or plant tissue area in pixels (**top panel**). Area was calculated using nCDA models and proper threshold value to include only relevant tissue and seedlings pixels. Sample of nCDA transformed image where every pixel is scored and only pixels above a certain threshold are counted (**middle panel**). Seedlings from all treatments were isolated and measured, blob (binary large object) collections with samples of these seeds and seedlings sorted by tissue/seedling area for seed lots 3 (dark blue circles) and 4 (light blue circles) are illustrated here (**bottom panels**).

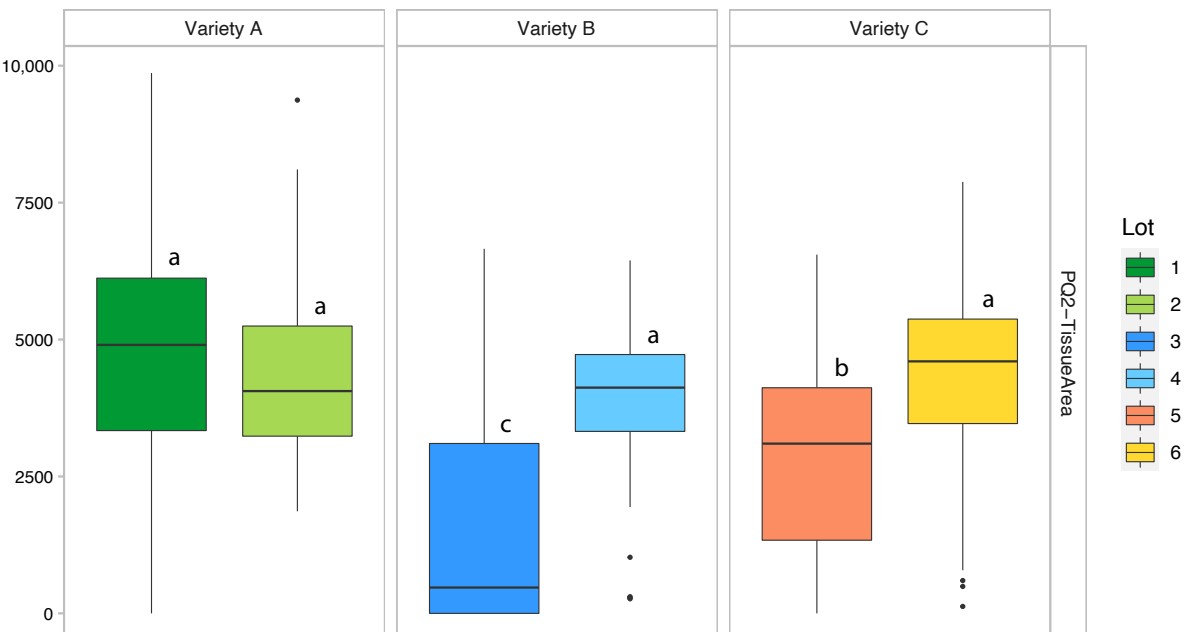

**Figure 9.** Seedling area (in pixels) of induced seeds after seed respiration measurements and before transfer to the greenhouse for plant growth and evaluation. Letters indicate significance difference among all lots as calculated by ANOVA and Tukey HSD.

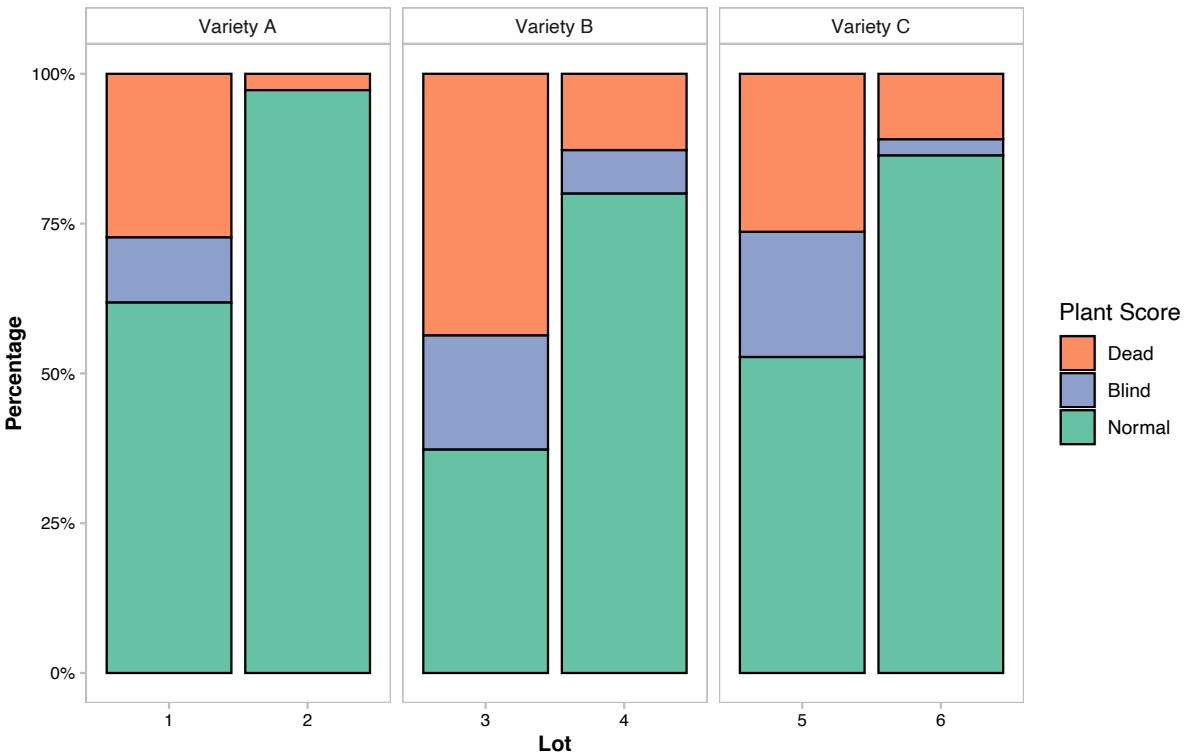

**Figure 10.** Plant evaluation for induced seeds and scores for dead, blind and normal seedlings after 2.5–5 weeks of growth in the greenhouse following 48 h at 1.5 °C and 72 h at 15 °C for Q2 measurements. Plants were scored per the descriptions in Table 1.

All individual seed data for chlorophyll content, multispectral reflectance, seed respiration and greenhouse plant evaluation scores were consolidated in one data file along with the variety, lot and repetition number (Supplemental Table S1). A full correlation matrix (Supplemental Figures S3 and S4) was constructed using all the data, enabling inspection of relationships among a large number of parameters at once to direct further analyses. To summarize the most critical information and avoid duplicating data, some primary parameters were selected and are presented in a smaller correlation matrix (Figure 11). The correlation numbers presented here provide an indication of their potential for use in individual seed sorting using the intersected parameters.

As expected, some parameters collected in the same instruments throughout all stages of the study were highly correlated with each other (Figure 11; Supplemental Figures S3 and S4, darker blue and red clusters). Correlations between the different analytical instruments used were also expected and observed in some cases. The chlorophyll fluorescence parameters from the CF-Analyzer were correlated with each other and also exhibited a strong correlation with multispectral parameters collected in the VideometerLab at different experimental stages (Figure 11, rows 1 and 2). This relationship was anticipated as both instruments are based on spectral imaging, with the CF-Analyzer targeting only chlorophyll content measurements with specific excitation wavelength and fluorescence wavelength filter while our VideometerLab version measures a broad range of wavelengths but lacks fluorescence filters (although addition of these is possible in the instrument).

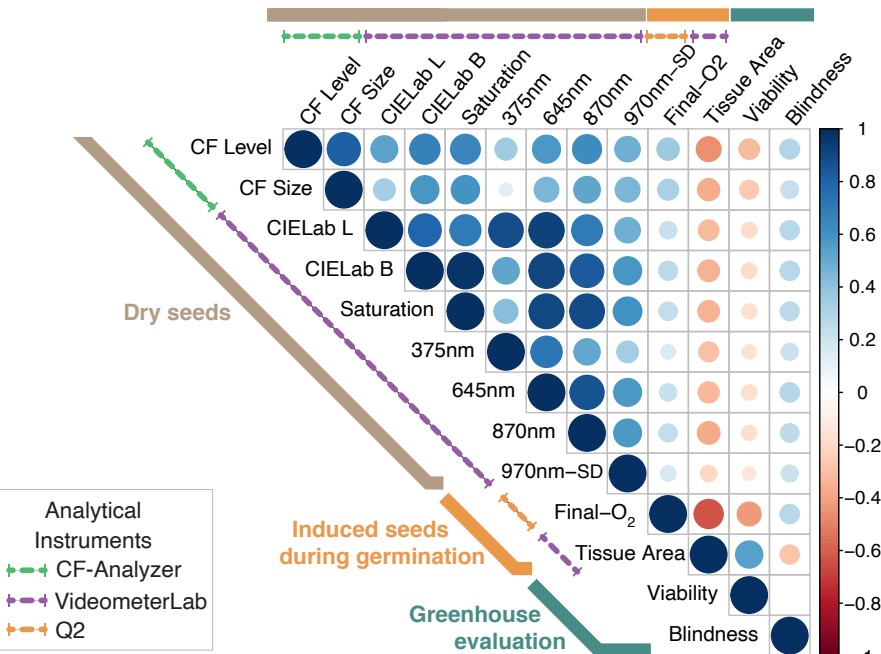

**Figure 11.** Pearson correlation matrix of selected main parameters (Table 1) from the CF Analyzer (rows and columns 1–2, dashed green), the VideometerLab (rows and columns 3–9 and 11, dashed purple) the Q2 respirometer (row and column 10, dashed orange) and greenhouse tests (rows and columns 12–13) for all induced seeds. Parameters are also organized in the order they were captured in the different stages of the study: dry seeds (rows and columns 1–9, brown bars), induced seeds during germination (rows and columns 10–11, orange bars) and in the greenhouse (rows and columns 12–13, green bars). The Pearson correlation coefficients between the row and column parameters are graphically displayed as circles; their size and color shadings indicate the strength of the correlation coefficients from 0 to 1 for a positive relation (blue color) or 0 to −1 for a negative relation (red color) (only correlation coefficients with significance level for p values below 0.01 are displayed).

The CF level (Figure 11, row 1 and all columns) had a significant relationship with VideometerLab color space parameters collected from dry seeds (CIELab L and B, r = 0.53 and 0.67, respectively, *p* < 0.001), saturation (r = 0.65, *p* < 0.001) and several wavelengths, including the ultraviolet (UV, 375 nm) and the near-infrared (NIR, 875 nm) ranges, with the highest correlation with 780 nm (r = 0.62, *p* < 0.001). This seed maturity indicator was also correlated (positively or negatively) to measurements performed at different stages of the study, such as multispectral imaging after the blindness induction (Supplemental Figure S3), during germination (final oxygen concentration in the Q2 and seedling area after the Q2, r = 0.37/0.45, respectively, *p* < 0.001) and with plant performance in the greenhouse (viability and blindness, r = −0.31/0.29, respectively, *p* < 0.001). Similar relationships were observed for the CF size parameter with weaker but still highly significant (*p* < 0.001) correlations for the dry seed and after induction, germination and greenhouse stages (Figure 11, row 2 and all columns).

The VideometerLab provides numerous parameters to quantify seed characteristics related to size, shape, spectra and others, but it also allows the application of customized features such as the tissue area and white spots/markings that we developed and used in this study (Supplemental Figure S4). In addition to the strong relationship with CF parameters, several of these features collected in dry seeds were correlated with VideometerLab features collected at different stages, Q2 measurements, and plant performance scores in the greenhouse. Some VideometerLab features collected at the dry seed stage were strongly correlated with data collected after blindness induction; these included the color space and saturation parameters from dry seeds and the percentage change in the same parameters after blindness induction (Supplemental Figure S3). The percentage change in the NIR

reflectance variation (PBI-970 nm-SD, Figure 4) was the PBI parameter with the highest (negative) correlation coefficients with Q2 parameters (Final-$O_2$, r = −0.38, *p* < 0.001), seedling area and plant performance (blindness, r = −0.19, *p* < 0.001) (Supplemental Figure S3). These Videometer parameters obtained after the blindness induction were, in most cases, highly correlated with original parameters from dry seeds (e.g., PBI-970 nm-SD with the dry seed 970 nm-SD) and exhibited lower correlations with seed respiration and plant performance. Thus, their relevance for sorting purposes was diminished and they were not included in the correlation matrix.

Videometer color space parameters and saturation values from dry seeds were further associated with the final oxygen measured in the Q2 (r values ranging from 0.23 to 0.27, *p* < 0.001), seedling area after the Q2 (r = −0.31 to −0.34, *p* < 0.001) and viability (r = −0.17 to −0.18, *p* < 0.001) and blindness (r = 0.27 to 0.28, *p* < 0.001) in the greenhouse. Average reflectance of several wavelengths collected in dry seeds also displayed strong correlations with the changes after induction but also with seed respiration, seedling area and plant performance. Some examples of the main wavelengths include UV (375 nm), red (645 nm) and NIR (870 nm) wavelengths that displayed associations with final oxygen level (r = 0.16 to 0.24, *p* < 0.001), seedling area after the Q2 (r = −0.29 to −0.38, *p* < 0.001), viability (r = −0.15 to −0.18, *p* < 0.001) and blindness (r = 0.21 to 0.28, *p* < 0.001). These wavelengths displayed a similar relationships with the quality parameters, but their correlation with CF level was somewhat distinct, with the 375 nm wavelength displaying a lower correlation (r = 0.35, *p* < 0.001) while the red and NIR wavelengths were more closely associated with CF level (r = 0.58 and 0.62, respectively, *p* < 0.001).

Respiration measurements in the Q2 also displayed associations with seedling area and plant performance. The oxygen percentage after 72 h (Final-$O_2$, Figure 11—row and column 10) was highly negatively correlated with seedling tissue area (r = −0.63, *p* < 0.001) and plant viability (−0.42, *p* < 0.001) and positively with blindness (r = 0.28, *p* < 0.001). These results reinforce that seed respiration is a good indicator for germination timing; seeds with a higher oxygen consumption rate also germinated earlier and had more time for seedling growth and development. Furthermore, the seedling tissue area was highly correlated with plant viability (r = 0.55, *p* < 0.001) and negatively with blindness (r = −0.28, *p* < 0.001).

The correlation among these selected traits across all seed lots (Figure 11) is mostly preserved when this dataset is analyzed separately within each seed lot or within each variety (data not shown), although the strength of correlations varied among lots and varieties. The only exception was found in the seed lots from variety A where the relationship between CF level and blindness was not present, likely due to the absence or limited number of seeds displaying the blindness phenotype (Figure 10).

A multiple factor analysis (MFA) with integrated parameters correlated to blindness from the different stages of the study was utilized to verify whether the distinct plant scores (normal, blind or dead) could be discerned (Figure 12; Supplemental Table S3). The MFA reinforced the complexity of distinguishing among these classes using an unguided approach, but it also revealed a clear trend, with blind and dead classes having considerable overlap but being clearly separated from the normal class (Figure 12, left panel). The first dimension (Dim1 accounting for 57.9% of the total variation, x axis Figure 12) distinguished the majority of blind and dead seeds from the cluster of normal seeds. The main parameters that contributed to this separation were the Videometer parameters 645 nm, saturation, 870 nm, 375 nm, CIELab L and A, 970 nm-SD, followed by CF level and size (Figure 12, right panel). The second dimension (Dim2 accounting for 14.3% of the total variation, y axis Figure 12) was most effective in distinguishing between the normal and dead classes. The top parameters in this dimension were the Final-$O_2$ and the tissue area (antagonistic), which contributed together more than 80% of the total dimension, followed by smaller contributions from the 970 nm-SD, CIELab A, CF level, 645 nm and the saturation (Figure 12, right panel).

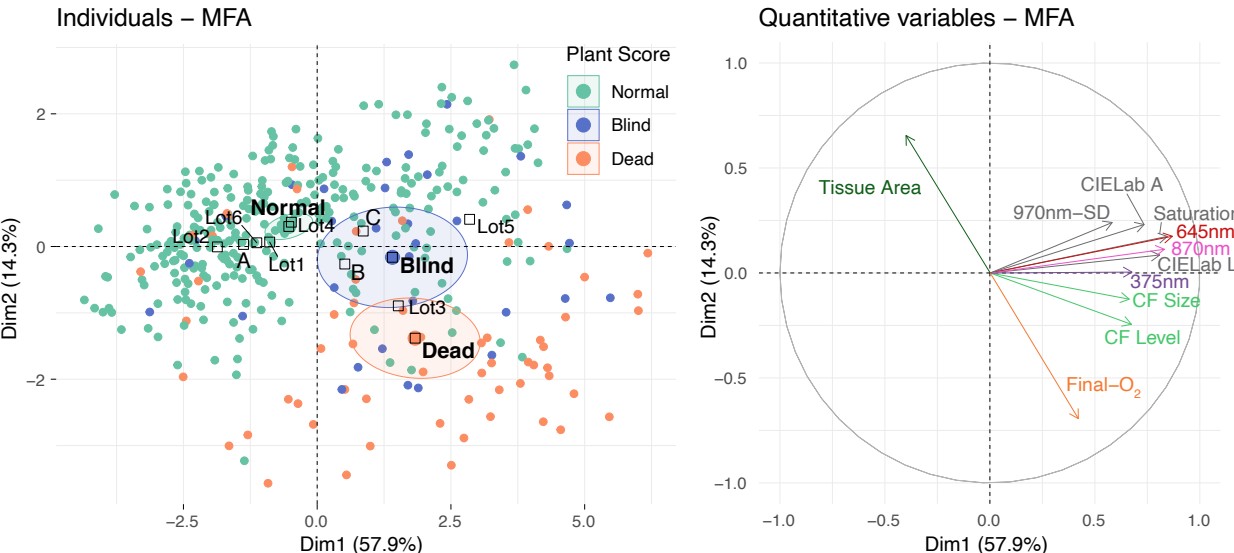

**Figure 12.** Multiple Factor Analysis (MFA) of features obtained from imaging, respiration and plant growth observations of kohlrabi seeds after blindness induction treatment. (**Left**): Scatter plot of individual seeds identified as normal, blind or dead separated in two dimensions (Dim1 and Dim2) according to their analytical features. Ellipses define confidence areas (95%) for each plant score, while squares represent their corresponding centers of gravity. Additional supplementary qualitative variables for seed variety (A, B or C) and lots (1 to 6) are shown in black. (**Right**): Vector representation of the influence of different measured factors in relation to their contribution to the two principal dimensions. Only complete observations for all parameters shown were used to generate the MFA.

## 4. Discussion

The potential for kohlrabi (and other *Brassica*) seeds to exhibit blindness, particularly after exposure to low temperatures, creates risks for both seed companies and growers. It has been difficult to identify the specific genetic and/or environmental factors during seed development that result in susceptibility to blindness. While conditions that can promote expression of blindness are known [1] and seed pretreatments can ameliorate susceptibility [3], it would be valuable to identify correlated traits that could be used for prescreening for blindness susceptibility to assign problematic lots for seed treatment or for sorting seed lots to remove the seeds that are susceptible to blindness. Thus, we examined both physical (CF-Analyzer, VideometerLab) and physiological (Q2) approaches to screening individual seeds prior to and after inducing blindness in order to identify whether it would be possible to predict which seeds or lots would be more likely to exhibit blindness.

All of the tested lots exhibited good initial performance at 20 °C, based on seed respiration time courses (Supplemental Figure S1). Lowering the temperature to 15 °C, however, resulted in much larger variances among seeds and discrimination among the seed lots (Supplemental Figure S2), with lots 2 (Variety A) and 4 (Variety B) exhibiting the greatest respiratory capacity at 15 °C (most seeds consuming most of the available oxygen). The subtle temperature stress resulted in few or no blind plants, so a blindness induction treatment was required to reveal the desired phenotype for this study. Following the induction treatment, lots 2 (Variety A) and 4 (Variety B) once more exhibited more active respiratory profiles at 15 °C (Figure 5), and also lower percentages of dead or blind seeds (Figure 10). In contrast, lots 1 (Variety A) and 3 (Variety B) showed greater impairment in respiratory activity at 15 °C and the highest susceptibility to blindness/death due to the induction treatment (Figures 5, 6 and 10). The behaviors of lots 5 and 6 from Variety C were intermediate, as these lots displayed a split respiratory behavior with about half of the seeds consuming most of the oxygen available while the other half consumed little to none (Figures 5 and 6). This result for lot 6 was rather anomalous, as it exhibited relatively poor respiratory capacity at 15 °C (Figure 5), but low susceptibility to blindness/death

(Figure 10). As the effect of the induction treatment increases progressively with longer times of exposure [1], it could be that lot 6 would show greater effects after a longer induction treatment. For the purposes of this experiment, the seed lots exhibited a range of susceptibility to death/blindness from the induction treatment, making it possible to test whether physical measurements would be related to this physiological behavior.

Differences among the seed lots at the dry seed stage were evident from parameters determined by the CF-Analyzer and the VideometerLab. Lots 3 (Variety B) and 5 (Variety C) exhibited higher CF values (Figure 2), as well as higher values for CIELab L, CIELab B, Saturation, 645 nm, 870 nm and the variation of the 970 nm reflectance (970 nm-SD) (Figure 3). The 375 nm and 435 nm values also identified lot 1 within the same group (exhibiting high values) as lots 3 and 5 (Figure 3), in agreement with these lots having more dead and blind seeds (Figure 10). Thus, the shorter wavelength parameters, specially 375 nm, suggest a new possibility for sorting, as that wavelength was not as highly correlated with CF Level as the 645 nm and 870 nm measurements (Figure 11). The 375 nm measures may detect another factor that could also be related to seed maturity. As higher values for all these measurements were associated with greater blindness and fewer normal seedlings (Figure 11), more immature seeds, indicated by higher chlorophyll levels, therefore appear to be associated with greater susceptibility to damage by low temperature imbibition.

Measurements performed after the low temperature induction treatment overlapped among seed lots (Figure 4). The saturation value change displayed significant relationships with blindness, but the more relevant relationship was observed with the NIR variation parameter (PBI-970 nm-SD) (Supplemental Figure S3), which was also somewhat more efficient to separate lots (Figure 4). This feature is derived from the variation in the NIR 970 nm reflectance (970 nm-SD) among dry seeds and exhibited a high correlation with that parameter. Both parameters separated out the three lots with higher blindness+death scores within varieties (lots 1, 3 and 5; Figure 3), suggesting that seeds with the larger initial variation could be linked to higher susceptibility to blindness, adding another signal option to aid sorting. The NIR 970 nm wavelength has been used as an indicator of water status in different substances [29,30], and it could be quantifying moisture distribution in seeds in this study, but further research is required to confirm this. Cracking or splitting of the testa precedes radicle emergence from *Brassica* seeds by a few hours [31], and this could be a factor that would add seed surface variation (e.g., exposure of seed tissues and contrast with the seed coat) as well as some moisture differences that could have been measured between active live and damaged or dead tissues.

Seed respiration during germination after the induction treatment also revealed differences among lots. Seed lots 1 (Variety A) and 3 (Variety B) were ranked with the lowest respiratory potential over several parameters (Figure 7, higher values), which agrees with their higher blindness+death scores (Figure 10). However, seed lots from Variety C (lots 5 and 6) usually had overlapping or inverted results when compared to their blindness+death scores (Figures 7 and 10). This issue can be better visualized with the POD curves (R75-POD Curves, Figure 6), as lot 5 has a fraction of faster respirators while lot 6 has a similar fraction of slower respirators; both lots had approximately 30% of seeds that did not consume 25% of the available oxygen (see also Figure 5). The use of POD curves provides a clear view of all seeds tested and avoids the selective calculation of averages and medians that do not account for seeds that did not reach a certain oxygen level. Additionally, a decreasing number of seeds is used to calculate the parameters when lowering the oxygen level threshold used. The final oxygen level or oxygen level at certain times can address this issue and show a realistic performance comparison at that time for all seeds tested. In this study, the final oxygen level (Final-$O_2$) was the Q2 parameter with stronger correlation with seedling area and plant performance in the greenhouse (Figure 11). This close connection between oxygen consumption and seedling area was expected and highlights the critical role of respiration in supporting early stages of plant growth [19]. Seed lots 1, 3 and 5 were ranked

as the lots with higher final oxygen levels in agreement with their blindness+death scores, but due to the larger overall variance present, seed lot 6 was also included in that group.

The seedling areas determined after the respiration measurements was also efficient in identifying lots 3 and 5 as presenting smaller seedling areas, and the large variation present in lot 1 (Figure 9). As expected, this parameter displayed a significant correlation with viability, with larger seedlings at the time of transplanting to greenhouse trays resulting in more viable plants. The relationship with blindness was also significant, although not as strong as with viability (Figure 11). The MFA analysis also shows how tissue area and Final-$O_2$ clearly separate dead and normal seedlings on opposite vectors (Figure 12).

To summarize the sorting opportunities at the seed lot level, mean values for chlorophyll, CIELab B, saturation, hue, NIR 870 nm and tissue area were able to distinguish lots with higher susceptibility to blindness in varieties B and C. The mean Q2 parameters (R75 and R50, Final-$O_2$) were able to separate the more susceptible lots of varieties A and B. Finally, mean values for CIElab L, 375, 435 and 645 nm, and 970 nm-SD were capable of differentiating these lots within all varieties. Most of these parameters displayed significant positive correlations and could be indicative for blindness susceptibility when high values are observed but also negatively correlated with the presence of normal seedlings. The MFA illustrates these contributions and the direction of separation, increasing with higher presence of blind seedlings and dead seeds while lowering towards higher frequency of normal seedlings (Figure 12, Dim1 both panels). On the contrary, tissue area had the opposite relationship with blindness and normal seedling percentages and a clear antagonistic relation with the Final-$O_2$ parameter (Figure 12 Dim2, both panels). While some of these parameters can be used to rank and sort all seed lots within these varieties, the importance of most parameters to identify blindness susceptibility seem to be variety-dependent. Sorting opportunities at the individual seed level may require larger sample sizes within lots and varieties to increase the pool of reference seeds displaying the phenotype of interest, expand the information available to properly account for the variation in seed physical characteristics and refine traits important for separation to provide higher confidence and accuracy.

## 5. Conclusions

The methodology and approaches used here demonstrate how a set of relevant parameters correlated to a phenotype of interest can be obtained using analytical instruments to assess individual seeds at different stages, starting from the dry seed through to the manifestation of the phenotype. The collection of the parameters from the different analytical instruments and/or stages combined can give valuable insight on how early or late these relevant parameters can be identified and used for seed lot management or upgrading. High-throughput equipment has been developed to physically separate individual seeds based on CF level (www.seqso.com (accessed on 13 January 2021)). Videometer A/S also recently developed a sorter with less speed and capacity, but capable of sorting seeds individually using multiple seed features or combinations of them. Additionally, new instruments and software are becoming available in which artificial intelligence is used to generate powerful algorithms to analyze seed images based on training sets (e.g., Seed-X, Magshimim, Israel). The procedure described here, of making digital images followed by assessing susceptibility to induction of blindness on a seed-by-seed basis, could be used for such training sets by identifying the greater or less susceptible seeds in the original images. At a minimum, the methods utilized here can efficiently identify lots with potential for injury or blindness in response to cold imbibition, which could then be processed further or pretreated to reduce their susceptibility. While further work is required to more fully confirm these approaches, the data provided here also demonstrate the possibility of sorting *Brassica* seed lots to remove individual seeds most susceptible to blindness following exposure to low temperatures.



**Supplementary Materials:** The following are available at https://www.mdpi.com/2077-0472/11/3/220/s1, Table S1: Seed parameters database, Table S2: Q2 parameters, Table S3: MFA Eigenvalues, Figure S1: Brassica control seed respiration curves at 20 °C, Figure S2: Brassica control seed respiration curves at 15 °C, Figure S3: Pearson correlation matrix of selected parameters, Figure S4: Full correlation matrix.

**Author Contributions:** Conceptualization, P.B. and K.J.B.; methodology, P.B. and K.J.B.; software, P.B.; validation, P.B. and K.J.B.; formal analysis, P.B.; investigation, P.B.; resources, P.B. and K.J.B.; data curation, P.B. and K.J.B.; writing—original draft preparation, P.B.; writing—review and editing, P.B. and K.J.B.; supervision, P.B. and K.J.B.; project administration, P.B. and K.J.B.; funding acquisition, K.J.B. All authors have read and agreed to the published version of the manuscript.

**Funding:** This research was funded by the Western Regional Seed Physiology Research Group.

**Institutional Review Board Statement:** Not applicable.

**Informed Consent Statement:** Not applicable.

**Data Availability Statement:** The work of the second author is supported by the Lomonosov Moscow State University under grant "Modern Problems of the Fundamental Mathematics and Mechanics".

**Acknowledgments:** The authors thank Corine de Groot for helpful suggestions, technical assistance and providing seed samples. We would like to acknowledge our colleagues Marlen Navarro Boulandier and Vincent Chiu for their initial experimental work and contribution to some of the original data presented here. Thanks to Peter Marks and Aginnovation for providing access to some analytical instruments utilized in these studies. We also thank Allen Van Deynze and Daniel Runcie for critically reviewing the manuscript prior to submission.

**Conflicts of Interest:** The authors declare no conflict of interest.

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
