# Peer review of "Relationships of Brassica Seed Physical Characteristics with Germination Performance and Plant Blindness"

_agriculture, doi:10.3390/agriculture11030220_

Round 1
Reviewer 1 Report
Overall well done and enjoyable to read.
The authors present an interesting paper on the development of screening protocols for the detection of blindness amongst Kohlrabi cultivars (Brassica oleracea) using high-throughput phenotyping methodologies. The paper is well organized and easy to read with little to no revision of the grammar and style required. There are some questions that arise while reading through the manuscript and some areas where clarification in methodology is required.
General questions:
- What is the incidence of blindness amongst Kohlrabi cultivars? In the authors first assessment only a single seed exhibited blindness. Have breeders made improvements on this trait and is it pleiotropic/linked to different cultivated phenotypes of Brassica oleracea? Given the economic impact stated by the authors some commentary on efforts to reduce blindness in Kohlrabi would be nice to see to tell a more complete story.
- Have the authors considered developing a predictive model for seed blindness, dead/dormant seed or normal seed? Similar to blindness, dead seed is another production challenge which is not always apparent prior to sowing. There are a number of highly correlated measurements taken within this experiment, some of which are redundant or exhibit slight correlation to the desired/undesired phenotype. The use of a dimension reduction techniques such as principal component regression may be on use here to aid in estimating the best predictors which could be applied to new seedlots to verify the findings presented here. This is the next natural step from my perspective.
Specific comments:
Line 61-63: Please provide further details on the maternal growth environment of tested seed lots. Where quality filtering parameters employed prior to receiving seed to reduce the incidence of blindness? The initial test observed only a single blind seed, are there alternative measures in place to reduce the incidence of blind seed in production fields?
Line 78: While excel is a commonly used spreadsheet software it should be referred to by its proper name Microsoft Excel version ## to avoid confusion.
Line 119: Different components of the experiments have two replicates, were there any instances of missing data? This will have implications in the downstream ANOVA.
Line 124: What were the lighting conditions in the greenhouse? Was this experiment run with natural or supplemental light? If using natural light please indicate the time of year to account for changes in photoperiod through the evaluation.
Line 124: Some Brassica species (including Brassica napus which shares the C genome with Brassica oleracea) exhibit secondary seed dormancy. Was seedlot viability assessed a priori or seed death verified by secondary methods such as tetrazolium testing? If not I suggest the authors replace the term “dead seed” with “dead and or dormant seed”. The authors state that oxygen consumption is normally associated with viable seed, what about viable but dormant seed? Please indicate if dormant seed is considered viable within the context of this study.
Line 130: Is normalized canonical discrimination analysis a function of the VideometerLab software packages? It is unclear if this was conducted in R or not. How was the data normalized? Was the data transformed, centered or scaled?
Line 136: Please define what type of correlation analysis was conducted: Pearson? Spearman? Kendall? Please ensure figures or tables referring to correlation results indicate what type of correlation is presented in the title/captions.
Line 136: Please state how assumptions of ANOVA were tested and met. Particularity of normality of residuals and if heterogeneity of variances were tested. Was there missing or unbalanced data?
Line 136: How were family-wise error rates accounted for? The correlation analysis includes a large number of significant but weak correlations. Are these all biologically significant? Many correlations are presented, I would suggest correcting for Type I errors or reducing dimensionality when presenting the results as this section is extremely dense.
Figure 11: This is a really visually interesting graphic but doesn’t add much to the story. Many of the measurements produced by the videometerlab device as highly correlated. This takes away the emphasis away from the more interesting correlations between seed lot characteristics and the various potential predictors. I think this figure may be better suited as a supplemental figure and a more condensed figure listing the phenotypes and their significant correlations makes for a clearer image.
General comments: I have no major issues with the paper in its current form with the exception of some missing details within the materials and methods. The discussion and conclusions appear to reflect the results appropriately. The paper in general is very dense and presents and tremendous amount of information, I feel it could be paired down somewhat and the presentation of traits which exhibit low correlations to the desired/undesired phenotypes could be reserved for the supplemental materials. Very interesting work, overall well done.
Reviewer 2 Report
Dear authors,
I believe this manuscript holds novel information. However, the way of presentation and readability can be greatly improved.
General:
*********
- minor textual corrections and suggestions are given in the attached document.
- ANOVA was used to assess possible differences between seed lots. However, the M&M does not describe if the normality of the distribution was checked. Various figures present a lot of outliers, which makes me suspect that observations are not always normally distributed. You should check this. If not normally distributed, ANOVA should not be used, but non-parametric tests such as Kruskal-Wallis
- Tukey and not Tuckey
- the paper gives a lot of information, but information is not given in a condensed way. Important findings are often 'hidden' in the middle of a text block, which makes it difficult for the reader. Sentences are generally long and complicated. When re-writing, try to restrict the amount of information given. Try to put important findings at the beginning or end of a text block.
Introduction
*************
Clear introduction and well written. What is missing are previous studies on the topic of blindness in cabbage. Are there any previous studies that have tried to predict blindness by studying seed characteristics or other traits? If not, it makes your study more novel and you should surely mention so.
M&M
****
I believe you should give the actual variety names, in case someone wants to repeat your experiments.
Pictures would make the methods used in this paper more clear. Can you present pictures for the blinded seedlings vs normal seedlings? Do you have nice pictures from your experiments, e.g. the respiration exp.?
Results
********
General remark, which applies for the whole results section:
- when stating whether a difference is significant or not, you must give the p-value within brackets (e.g. line 164-166) or indicate the significance with ***.
- when stating x is higher than y, you should give the actual values between brackets, e.g. line 167-168: "Seed lots 5 (C), followed by 3 (B) displayed higher median chlorophyll contents (... and .. respectively) when compared to the other lots (...)" In the later part of the results, this is done well (e.g. line 306-307).
- line 209-211: this is repeated from the introduction, can be removed.
- line 211-218: this information should be in the M&M section. You should restrict it to max. 1 introductory sentence here.
- line 232: what do you mean by "negative variation"? In my opinion, there either is (positive) variation or no variation.
- line 253: reference should be given as [...]
- line 259: "slower rates" of what?
- line 286-294: this information belongs in M&M
- line 329: entire text block: what do the **** mean? I guess the significance levels/p-values, but it is not explained. I prefer giving the actual p-values instead of the *** code.
Discussion
***********
- Are there any previous studies on the same topic with which you can compare your results?
- line 439: "top scorer group": do you mean the seed lots least prone to blindness?
- As stated before, important findings are presented in a rather scattered way. A conclusion part would help the reader to easily see your major findings. I would add a short conclusive paragraph at the end of the discussion summarizing the major findings of this study, and what they mean to the general public. Which seed characteristics can the reader use to estimate the seeds' susceptibility to blindness?
Figures
********
- All figures: clarify in the legend that the columns A, B, C indicate kohlrabi varieties A, B and C. Indicate that you present the results for 3 kohlrabi varieties and 6 seed lots.
- Fig 1: Legend is not clear. Which plot is which variety?
- Fig 3: legend: it is not necessary to say alpha < 0.05: tin the M&M you already indicate that in all statistics you use alpha < 0.05. Please check every legend.
- Fig.11: what is presented on the y-axis? Last sentence of the legend: I think you mean "p-values above 0.05". This figure is not clear to me.

Author Response
Please see attachement file
